# Parental Competence and Pornography Use among Hispanic Adolescents

**DOI:** 10.3390/bs14100926

**Published:** 2024-10-10

**Authors:** María Calatrava, Paola Alexandria Pinto de Magalhaes, Marta Vidaurreta, Sonia Rivas, Cristina López-Del Burgo, Maider Belintxon

**Affiliations:** 1IdisNA, Navarra Institute for Health Research, Calle de Irunlarrea, 3, 31008 Pamplona, Navarra, Spain; mcalatrava@unav.es (M.C.); mvidaurreta@unav.es (M.V.); cldelburgo@unav.es (C.L.-D.B.); mbelintxon@unav.es (M.B.); 2Institute for Culture and Society, Campus Universitario, Universidad de Navarra, 31009 Pamplona, Navarra, Spain; 3School of Education and Psychology, Campus Universitario, Universidad de Navarra, 31009 Pamplona, Navarra, Spain; srivas@unav.es; 4Department of Community, Maternity and Paediatric Nursing, School of Nursing, Faculty of Nursing, Campus Universitario, Universidad de Navarra, 31009 Pamplona, Navarra, Spain; 5Clínica Universidad de Navarra, Avenida Pío XII, 31008 Pamplona, Navarra, Spain; 6Preventive Medicine and Public Health, School of Medicine, Campus Universitario, Universidad de Navarra, 31008 Pamplona, Navarra, Spain

**Keywords:** adolescents, pornography, parental competence, positive youth development, lifestyles

## Abstract

Objectives: This study aims to determine the association between parental competence (warmth, demandingness, and parental education in fortitude and in privacy) and pornography use. Methods: This study presents cross-sectional data from an ongoing international study (YOURLIFE Project) about the opinions and lifestyles of adolescents with respect to affectivity, love, and sexuality. Adolescents (N = 2516) aged 12–15 from Chile, Mexico, Peru, and Spain were included. Multiple logistic regressions were performed to analyze the association between family assets (parental demandingness, warmth, and parental fortitude and privacy education) and pornography use. Results: The results indicated an association between parental warmth and pornography use among boys and girls. Furthermore, privacy education was highly associated with less pornography use only in girls. Parental demandingness and parental fortitude education were not associated with pornography use. Conclusion: Our findings suggest that new educational perspectives including privacy issues should be considered within programs for pornography use prevention among adolescents. Parents should incorporate these variables when discussing pornography with their adolescents and not only focus on filters or demandingness.

## 1. Introduction

Adolescence is a stage of transition to adulthood, characterized by biological, psychological, and social changes that may promote the adoption of new behaviors and lifestyles that carry risks for teens’ health [1,2].

The most prevalent risk behaviors among young people are alcohol use [1,3], tobacco use ([4]—Bendaou et al., 2018; [5]—Nejjari et al., 2009), sexual risk behaviors [6,7,8] and “problematic smartphone use” [9]. This represents a major public health issue as it carries negative consequences for young people’s development and the health of the population in general [10].

The use of digital media has increased exponentially in the last decade [11,12], particularly among young people. Today, the vast majority of adolescents (95%) have their own smartphone with an internet connection [13]. The time adolescents dedicate to connecting to the internet has doubled between 2006 and 2016, reaching an average of six hours per day spent communicating via social media in their free time [14].

Recently, the internet has become the leading platform for pornography use [15,16]. The theory of triple A, i.e., access, affordability, and anonymity, has facilitated the consumption of pornography. The ease of access, the free availability of a huge quantity of content, and the anonymity of being behind a screen has increased adolescents’ use of pornography [17].

Studies on the use of pornography establish its prevalence as between 27 and 70.3% [18,19], with boys being more likely to consume this type of content than girls [20,21]. The age range for beginning to use pornography is located between 12 and 17 years [21], although some studies indicate that children are accessing pornography at increasingly earlier ages, situating their initial viewing at 8 years of age [22].

The use of pornography has been associated with early sexual activity [23,24]; violent sexual behavior, particularly towards women [25,26]; the practice of sexting (sending sexually suggestive images or videos to one another) [27,28,29]; and viewing women as sexual objects [28,29]. It has also been linked with greater physical insecurity regarding one’s own body [30,31,32], lower self-esteem [33], poor mental health [34] and the deterioration of attachment bonds, which may result in more dysfunctional relationships and social isolation [34,35,36].

Parents seem to play a modulating role in their children’s sexual behaviors [37] and may exercise a protective role with regard to the use of online pornography. Parenting styles and parenting competence are crucial in understanding the dynamics of parent–child relationships and their impact on child development [38].

Parenting styles are defined by parental behavioral patterns that aim to control and socialize children, establishing an emotional context for parent–child relationships [38]. Reparaz et al. [39] consider parenting styles as a contextual variable that moderates the relationships between the parents’ specific parenting practices and the children’s development outcomes. Prior studies indicate that warm and close family relationships between parents and children, together with an authoritative parenting style that combines high parental control with high emotional availability, protect against the use of pornography in adolescents [40,41]. In contrast, an authoritarian parenting style, characterized by high parental monitoring but little affection, seems to increase this [42]. On the other hand, insufficient affection from parents may encourage insecure attachment in adolescents, resulting in an increased use of online pornography and encouraging other sexual risk behaviors as an emotional compensation mechanism [42,43]. Empirical studies often focus on how parents’ educational styles promote online pornography consumption, ignoring the role of parenting competence.

Parental competence is defined as the parents’ practical abilities that they use to protect and educate their children, and to guarantee their healthy development [44]. More specifically, it refers to the ability of parents to perform their role effectively, and it encompasses attributes such as self-regulation, self-esteem, promotion of children’s self-esteem, conflict resolution, and communication [45]. The parental competence concept includes parenting styles (demandingness and warmth) and education in values (fortitude and privacy).

Reparaz et al. [39] and Burke et al. [46] show that education in fortitude and privacy could be protective against addictions. Education in fortitude refers to the psychological strength of youth and their ability to manage adversity and promote self-wellbeing [47]. Fortitude is defined as the strength to cope with stress and stay well and arises from positive cognitive appraisals of the self (e.g., as competent and capable), the family (e.g., as reliable, trustworthy, and supportive) and external sources of support (e.g., as available, accessible, and potentially beneficial) [48]. Privacy education means teaching young people the value of their own intimacy and that of others [49], which encourages children to better identify any content that does not correspond to this value and communicate this to their parents [50]. No prior studies have studied the relationship between these variables and the use of pornography.

The importance of parental styles in parental competence regarding pornography exposure is obvious, however, no empirical research has been found to date that investigates parental education in fortitude and in privacy, as an aspect of parental competence, and its relationship to pornography exposure.

Furthermore, in a recent literature review about pornography use and adolescents, more than two-thirds of the articles reviewed came from Europe, North America, or Australia, with no articles including countries located in Central or South America [51]. Research shows that living in a more liberal country predicted a higher probability of intentional exposure to sexually explicit online materials [52]. Also, culture has a significant impact on the type of family relationships or the type of parental styles that a family adopts [53]. Therefore, it seems scientifically relevant to show how these associations occur, taking into account cultural differences. We propose the examination of the relationship between parental competence and the use of pornography among adolescents in Spanish-speaking countries to compare with studies from other cultures.

For this, the following objectives have been set:(1)To describe the characteristics of adolescents who use pornography in a sample of four Spanish-speaking countries (Spain, Chile, Mexico, and Pero), according to sex differences.(2)To evaluate the relationship between parental competences (warmth, demandingness, and parental education in fortitude and in privacy) and the use pornography among our study participants by sex.

## 2. Materials and Methods

This research forms part of an ongoing international study called the YOURLIFE Project (https://proyectoyourlife.com/en/ accessed on 29 September 2024), the main objective of which is to discover adolescents’ lifestyles and identify what they think and feel about love, sexuality, and partner relationships, as well as the factors that influence them [53,54]. For this study, cross-sectional data has been analyzed from four Spanish-speaking countries (Chile, Spain, Mexico, and Peru).

### 2.1. Participants and Sampling

A convenience sample was obtained by disseminating information about the study across secondary education centers—both public and private. A total of 52 schools voluntarily reached out to participate, and an online self-administered questionnaire was completed by 3443 secondary school students (12–15 years old). Those participants who did not indicate their age or sex were eliminated (n = 143). In addition, those that did not answer the main outcome (n = 284) and the four parental competences (n = 500) were also excluded from analyses. Consequently, a total of 2516 students were included in the final analysis (288 from Chile, 731 from Spain, 842 from Mexico, and 655 from Peru), representing 73.1% of the initial sample.

### 2.2. Questionnaire and Variables

The YOURLIFE project questionnaires were used to collect data. The YOURLIFE project is composed of three questionnaires (Q13, Q15, and Q17) administered according to the age of the study population. The data for this study were collected using the Q13 questionnaire, specifically designed for adolescents aged from 12 to 15. Notably, the questions addressing parenting styles—central to our research—were exclusive to this wave, meaning that only participants within this age range provided responses. As a result, data from older adolescents were not available for analysis. These questionnaires have been previously described in other studies [53,55]. This is an anonymous, self-administered, online questionnaire regarding adolescents’ lifestyles and interpersonal relations. The variables used in this study are described below.

### 2.3. Outcome: Use of Pornography

Participants were asked the frequency of their use of digital pornographic content (never, less than one day a month, between 1 and 3 days a month, 1–2 days a week, or 3 or more days a week). Responses were coded into a new dichotomous variable (yes/no), giving a 0 to those who had never used pornographic material and a 1 to all other use frequencies. Missing data were eliminated.

### 2.4. Main Independent Variables: Parental Competences

Parental competence was measured through the Family Education Scale (FES). This instrument assesses parenting styles (demandingness and warmth) and education in values (fortitude and privacy) and has been validated in Spanish-speaking countries [39].

### 2.5. Parental Warmth

The questions regarding parental warmth were formulated as follows: Regarding your parents: “Do they know you well and understand you?”; “Do they set an example?”; “Do they listen to you?”; “Do they take your opinions into account when doing something?”; “Do they speak kindly to you?”; “Do they help you when you feel insecure?”; “Do you feel that they love you and that they accept you as you are?”; “Do you feel comforted and supported by them?”; “Do you feel that your things interest them?”; “Do they take time to talk to you?”; and “Do they try to be with you and help you?”. Each question had five possible answers (from 0 = not at all to 4 = very much). The mean score for each participant was calculated based on the items they had completed, as long as they had responded to at least half of the items. Participants who answered fewer than half of the items were excluded from the analysis. This variable was dichotomized around the median into “high warmth” and “low warmth”. Internal consistency was high (Cronbach’s alpha = 0.95).

### 2.6. Parental Demandingness

The questions regarding parental demandingness were formulated as follows: Regarding your parents: “Do they require you to follow a schedule?”; “Do they decide with you what you must do?”; “Do they limit what you spend?”; “Do they limit the time you can watch television?”; “Do they control your use of cell phones or the Internet?”; and “Do they control your books and magazines?”. Each question had five possible answers (from 0 = not at all to 4 = very much). The mean score was calculated for each participant based on the items they had completed, provided they had answered at least half of the items. Participants who responded to fewer than half of the items were excluded from the analysis. This variable was dichotomized around the median into “high demandingness” and “low demandingness”. Internal consistency was acceptable (Cronbach’s alpha = 0.74).

### 2.7. Education in Fortitude

The questions set to adolescents to discuss education in fortitude were formulated as follows: “do your parents teach you to see the positive side of things?”, “do they teach you not to worry about things?”, “do they teach to reject whims?”, “do they teach you not to do something just because others are doing it?”, “do they teach you to improve and achieve your goals?”, “do they teach you to express your opinion?”, “do they teach you to defend your ideas?”, “do they teach you to listen to others ideas?”. Each question had five possible answers (from 0 = never/not at all to 4 = always/very much). The mean score was calculated for each participant based on the items they had completed, provided they had answered at least half of the items. Participants who responded to fewer than half of the items were excluded from the analysis. This variable was dichotomized around the median into “high education in fortitude” and “low education in fortitude”. Internal consistency was high (Cronbach’s alpha = 0.90).

### 2.8. Education in Privacy

The following questions were included in order to discuss education in privacy: “do your parents encourage you not to share your problems and feelings with untrustworthy people?”, “do your parents encourage you to be careful in how you dress to avoid making others uncomfortable?”, “do your parents encourage you to avoid looking at images or listening to songs with sexual content?”, “do your parents encourage you to value your body’s intimacy?”, “do your parents encourage you to look after your physical appearance?”, “do your parents encourage you not to obsess over your physical appearance?”, “do your parents encourage you to avoid lying and pretending in chats and on social media?”, “do your parents encourage you not to share personal information (yours, your family’s or your friends) to other people online?”, “do your parents encourage you not to take photos, save conversations or publish other people’s things online without permission?”, “do your parents encourage you not to speak in public about things you know regarding friends and family?”. Each question had five possible answers (from 0 = never/not at all to 4 = always/very much). The mean score was calculated for each participant based on the items they had completed, provided they had answered at least half of the items. Participants who responded to fewer than half of the items were excluded from the analysis. This variable was dichotomized around the median into “high education in fortitude” and “low education in fortitude”. Internal consistency was high (Cronbach’s alpha = 0.90).

### 2.9. Covariates

The multivariate analyses were adjusted for sex, educational level of the parents, having a parental content filter at home, owning a smartphone/tablet with mobile data, and religiosity.

The presence of parental content filters at home was measured as a single item with three response categories (yes/no/don’t know). Those who responded that they did have a parental content filter were assigned to category 1. Those who stated that they did not, or that they did not know, were assigned to category 0. Subjects that did not answer this question were assigned to category 0.

The variable of owning a smartphone/tablet with mobile data was also included, with two response categories (yes/no). Subjects who did not answer this question were included in the “no” category.

Levels of religiosity were analyzed through four items. Students were asked what religion they belonged to. Then, if they belonged to any religion, they were asked how often they went to the church/temple of their religion and how often they prayed (from 0 = Never, to 5 = More than once a week). They were finally asked how much they agreed with this statement: “My faith is an important influence in my life, and I am willing to take it into account in my decisions” (from 0 = strongly disagree to 4 = strongly agree). From all these items, a dichotomous variable called religiosity was generated into “high religiosity” and “null/low religiosity”. “High” religiosity therefore included adolescents who had a religion, went to church weekly, prayed at least weekly, and agreed or strongly agreed that their faith was an important influence. The rest of the participants were included in the category of “none/low religiosity”.

### 2.10. Data Collection

The study was conducted in four countries: Chile, Spain, Mexico, and Peru. Schools were invited to participate through various channels, including email, personal contacts, and a network of local partners. Schools that expressed interest received a detailed protocol outlining the data collection process. Students completed the questionnaires during class time in the schools’ IT rooms. Each school was responsible for managing parental consent according to local laws and policies, following the guidance provided by the research team [56]. Depending on the protocol chosen by each school, consent could range from implicit parental authorization to explicit consent. Students participated voluntarily, and they were informed that they could withdraw from the survey at any time or skip any question. Throughout the process, students’ privacy was strictly protected, and no identifying information was linked to their responses.

### 2.11. Data Analyses

The participants’ characteristics were indicated as absolute frequencies and in percentages, according to sex. Firstly, bivariate analysis was carried out in order to describe and compare the use of pornography in boys and girls with respect to the variables (warmth, demandingness, privacy and fortitude education, parents’ educational level, owning a smartphone/tablet with mobile data, having a parental content filter installed in the home, and religiosity). Because a preponderance of the adolescent participants reported that they had never used pornography, aggregate scores for these scales were dichotomized (0 = absent; 1 = present).

Models of gender-stratified, non-conditional multivariate logistic regression models were carried out to evaluate the link between family, as an external asset, and the use of pornography. Firstly, we created a model for each family characteristic, adjusting for sex, country, religiosity, owning a smartphone/tablet with mobile data, having a content filter at home, and the parents’ educational level. Subsequently, all family characteristics were introduced into a single model, adjusting for the rest of the variables. 

All analyses were carried out with the Stat V.12. statistical program, with a significance level of 0.05.

### 2.12. Ethical Considerations

The project was approved by the Ethics Committee of University of Navarra.

## 3. Results

### 3.1. Characteristics of the Participants

Table 1 shows the main characteristics of the participants. The sample mainly consisted of girls (56.7%) and the predominant age was 13 years (54.9%). Sixty-six percent of adolescents surveyed had at least one parent who had undertaken university studies. The majority of the participants (78.7%) declared themselves as Catholic. All family variables studied (warmth, demandingness, education in privacy and education in fortitude) presented with a homogenous distribution among the participants of our sample.

### 3.2. Sex Differences in Pornography Use

The differences in pornography use by sex are presented below and detailed in Table 2.

Among boys, higher parental warmth was associated with lower levels of pornography use (OR = 0.64, 95% CI: 0.48–0.85), as was higher parental demandingness (OR = 0.66, 95% CI: 0.50–0.87). Boys with higher scores in education in privacy also reported less pornography use (OR = 0.63, 95% CI: 0.47–0.85). Although parental education in fortitude showed a trend towards significance (OR = 0.75, 95% CI: 0.56–1.00), it was not statistically significant. High religiosity was associated with lower pornography use (OR = 0.63, 95% CI: 0.46–0.85). On the other hand, boys with parents who had a university education (OR = 1.50, 95% CI: 1.08–2.09), those with a smartphone/tablet with mobile data (OR = 1.79, 95% CI: 1.32–2.42), and those without parental content filters (OR = 0.58, 95% CI: 0.40–0.86) reported higher levels of pornography use.

For girls, parental warmth (OR = 0.36, 95% CI: 0.22–0.60) and parental demandingness (OR = 0.52, 95% CI: 0.32–0.84) were strongly associated with lower levels of pornography use. Girls with higher scores in education in privacy (OR = 0.33, 95% CI: 0.20–0.54) and education in fortitude (OR = 0.51, 95% CI: 0.31–0.82) also reported significantly lower use. High religiosity was similarly associated with reduced pornography use (OR = 0.48, 95% CI: 0.27–0.85). Other variables, such as parental university education, smartphone/tablet ownership, or the presence of parental content filters, did not show significant associations with pornography use among girls.

### 3.3. Multivariate Analyses

According to the multivariate analyses (Table 3, see the bottom of the article), high levels of parental warmth were associated with reduced pornography use in both boys (OR = 0.70, 95% CI: 0.50–0.98) and girls (OR = 0.45, 95% CI: 0.25–0.80). Additionally, higher scores in the education in privacy subscale were linked to reduced pornography use among girls (OR = 0.45, 95% CI: 0.26–0.77). Although the univariate analyses showed that parental education in privacy was significantly associated with lower pornography use in boys (*p* < 0.05, Table 2), this association was only marginally significant in the multivariate analysis (OR = 0.76, 95% CI: 0.54–1.07).

For boys, high religiosity was associated with lower pornography use (OR = 0.66, 95% CI: 0.48–0.91). In contrast, factors associated with greater pornography use included being a boy from Chile (OR = 2.04, 95% CI: 1.31–3.20), having at least one parent with a university education (OR = 1.70, 95% CI: 1.20–2.39), and the availability of mobile data on a personal device (OR = 1.80, 95% CI: 1.32–2.46). Education in fortitude and the presence of parental content filters at home were not significant in the multivariate analysis.

## 4. Discussion

The aim of this study was to evaluate the association between certain parental competences and the use of pornography content in a sample of Spanish and Latin American adolescents.

Regarding the prevalence of pornography consumption among secondary school students (12–15 years-old), our study is in line with recent studies that demonstrate pornography consumption among secondary school students [57,58] and that they are exposed to it at a younger age [59]. Authors have shown that the youngest age at which children are being exposed to pornography is from 10 to 11 years old for boys, and 12 to 13 for girls [60]. A probable explanation for this is the easy access to both smartphones and pornography, with the smartphone facilitating access through the internet [60].

In the same line as other studies in the field of pornography, boys consume more frequently pornography than girls [61]. Studies suggest that this is due to the fact that, for men, access to pornography is more acceptable [62], and men have a more positive attitude toward pornography than women [63,64]. Boys tend to use it more frequently and for sexual arousal, while girls may use it for the exploration of gender boundaries and sexual identity [65].

Regarding parental competences, those adolescents that indicated having received more warmth from their parents reported less use of pornography. These results are consistent with other studies carried out in different countries. In a sample of Chinese adolescents, it was found that higher levels of reciprocity (support, love, and concern) were associated with lower levels of pornography use [66,67]. According to the self-determination theory, parental support for the needs of relationships, autonomy, and skills is crucial in order to achieve optimal psychological growth and health [68]. Adequate parental bonds may help self-regulation and reduce dependence on digital media [69]. On the other hand, adolescents with a lack of warmth in their family presented with a higher use of online content, including pornography—as a potential compensation mechanism for their emotional needs [43].

Likewise, our study has also underlined that family communication (“Do they speak kindly to you?”; “Do you feel that they are interested in you?”) may play an important role in the transmission of warmth as a protective factor against this use. In accordance with other authors, when parents generate a family climate which transmits warmth through communication, children are more likely to develop greater self-esteem and demonstrate less perceived stress [33,70]. It should be noted that teenagers with low self-esteem used pornography more frequently [71]. Similarly, parents with an educational style characterized by a greater degree of warmth were more likely to understand the seriousness of pornography use in their children, appearing to be more active in the three central strategies of parental mediation: restrictive (rule setting for media use), active (parents discussing media content with their children), and shared use (shared media time with children) in relation to the use of pornography [52]. In summary, a warm relationship between parents and children seems to encourage children to develop protective skills against pornography use.

With respect to education in fortitude, no relation was found with the use of pornography in our sample. Our initial hypothesis was that education based on messages that encourage young people to grow in fortitude could have beneficial effects on internet use. However, it has been established that adolescents that show high levels of fortitude had lower levels of trauma symptoms regarding exposure to violence [48]. The promotion of fortitude is considered as a positive factor in increasing an individual’s capacity to resist pressures and stressors that may emerge at any point in their lives [71]. In adolescence, this pressure may come from the peer group, as peers play an important role in influencing the behavior of adolescents [72]. As such, parental education in fortitude is essential, as it has been confirmed that peer pressure is related to pornography use [20,51,73,74].

The results of our study also demonstrate that greater levels of education in privacy correlate with lower levels of pornography use in adolescents—particularly among girls. The majority of prior studies carried out have only considered parental education in privacy as the set of messages aimed at warning minors of the risks of sharing personal information online [75]. This differs from the current study, which includes a broader and more comprehensive perspective for assessing if adolescents receive messages from their parents regarding “valuing the intimacy of their body” or “not speaking in public about things related to other people”. Today, online content strongly focused on sexual appearance, physical beauty, and sexual attraction for others is prevalent [76]. This seems to form part of a social phenomenon known as “social extimacy” [77,78,79], characterized by the presence of excessive individualism online and greater online visibility of content that should belong to the personal sphere. Having this ability to discriminate is even more relevant during adolescence, when peer relationships take a central role in identity development [80].

Although our results suggest a possible protective effect of parental demandingness over the use of digital pornographic content, this was not significant. Our findings are consistent with other studies that did not find an association between demandingness and use. Adolescents are able to evade the supervision of their parents and connect to the internet from places where they cannot be controlled [41,51,81,82,83]. On the other hand, parental control tools are essential for regulating internet usage and mitigating potential harms [84,85]. These kinds of tools can help parents monitor and control online activities, including blocking inappropriate content [86,87].

### Limitations and Strengths

This study presents certain limitations. Firstly, the results have been derived from a cross-sectional study that does not allow for causality to be inferred regarding the observed associations. Secondly, we have used a convenience sample. However, this problem may have been partially offset by our large sample size, which facilitated the proper adjustments for several potential confounders. Thirdly, pornography use was measured solely based on frequency, and the participants were categorized into two groups: users and non-users. Similarly, other key variables, such as parental warmth and demandingness, were dichotomized into high and low categories using median splits. While this approach provides a general overview, it may reduce the study’s statistical power and limit the granularity of the analysis. Future research should explore more nuanced categorizations or continuous measurements to capture a broader range of behaviors and characteristics. On the other hand, the issue of whether this use of pornography was intentional or unintentional was not considered. Despite this, our results showed that the parental competences reported protected against this use, independently of this. Future studies should consider this aspect of use and whether or not both forms share the same risk and protection factors.

Despite these limitations, this study expands upon existing findings by showing that another factor, parental education, is associated with pornography, beyond the frequently studied variables of demandingness and warmth, thus providing a more accurate analysis for each family dimension and clarifying the role of the family. Furthermore, the study was conducted within a Hispanic context—an insufficiently explored population—addressing the gap identified by Peter and Valkenburg [51].

## 5. Conclusions

The results from this cross-sectional study confirm that family education based on warmth and education in privacy protects Hispanic adolescents from using pornography. Parenting styles and parenting practices regarding the use of pornographic content are still key elements that may have an effect on its consumption. Not only should parents pay attention to well-known parental practices such as controlling the use of cell phones or the Internet, but they should also note that education in privacy is another important but lesser known factor. The key messages and practical implications of this research include the following:Parents should continue to build caring relationships with their children.Parents’ communication with adolescents should include discussions about how they can determine whom to trust, teaching their teen to value their body´s intimacy, not to obsess over their physical appearance, and not spreading gossip about friends and family.These aspects should be considered when addressing pornography consumption through adolescent and parental programs.Encouraging and educating adolescents to protect their intimacy could be useful in reducing pornography use and improving an adolescent’s development from early in the adolescent period.Other variables, including education in fortitude, should be further investigated.

## Figures and Tables

**Table 1 behavsci-14-00926-t001:** Characteristics of the sample.

Characteristics	Boys n (%) N = 1090 (43.3)		Girls n (%) N = 1426 (56.7)		Total n (%) N = 2516	
**Age (years)**						
12	102	(9.4)	79	(5.5)	181	(7.2)
13	548	(50.3)	834	(58.5)	1382	(54.9)
14	361	(33.1)	441	(30.9)	802	(31.9)
15	79	(7.2)	72	(5.1)	151	(6.0)
**Country**						
México	409	(37.5)	433	(30.4)	842	(33.5)
Chile	123	(11.3)	165	(11.6)	288	(11.4)
Spain	301	(27.6)	430	(30.1)	731	(29.1)
Perú	257	(23.6)	398	(27.9)	655	(26.0)
**Parents with university education**						
No	313	(28.7)	543	(38.1)	856	(34.0)
Yes	777	(71.3)	883	(61.9)	1660	(66.0)
**Religion**						
No religión	170	(15.6)	252	(17.7)	422	(16.8)
Catholic	861	(79.0)	1119	(78.5)	1980	(78.7)
Other ^a^	59	(5.4.0)	55	(3.8)	114	(4.5)
**Use of pornography**	256	(23.5)	81	(5.7)	337	(13.4)
**Parental warmth**						
Low	535	(49.1)	744	(52.2)	1334	(53.0)
High	555	(51.0)	682	(47.8)	1182	(47.0)
**Parental demandingness**						
Low	529	(48.5)	799	(56.0)	1273	(51.0)
High	561	(51.5)	627	(44.0)	1243	(49.0)
**Education in the fortitude**						
Low	659	(60.5)	751	(53.0)	1410	(56.0)
High	431	(39.5)	675	(47.0)	1106	(44.0)
**Education in privacy**						
Low	679	(62.3)	670	(47.0)	1349	(53.6)
High	411	(37.7)	756	(53.0)	1167	(46.4)

^a^ Other religions include Reformed/Evangelical, Orthodox, Islam, Buddhism, Hinduism, Jewish, Traditional Chinese religion, and “Other”.

**Table 2 behavsci-14-00926-t002:** Pornography use in boys and girls by parental and personal variables.

Boys					Girls			
	*Total* *N = 1090*	*Porn Use* *n = 256 (%)*	*p* ^a^	*OR (95% CI)* ^b^	*Total* *N = 1426*	*Porn Use* *n = 81 (%)*	*p* ^a^	*OR (95% CI)* ^b^
** *Family education variables* **								
**Parental warmth**								
Low	529 (48.5)	146 (27.6)		(ref)	744 (52.2)	60 (8.1)		(ref)
High	561 (51.5)	110 (19.6)	<0.05	0.64 (0.48–0.85)	682 (47.8)	21 (3.1)	<0.001	0.36 (0.22–0.60)
**Parental demandingness**								
Low	535 (49.1)	146 (27.3)		(ref)	799 (56.0)	57 (7.1)		(ref)
High	555 (51.0)	110 (19.8)	<0.05	0.66 (0.50–0.87)	627 (44.0)	24 (3.8)	<0.05	0.52 (0.32–0.84)
**Education in fortitude**								
Low	659 (60.5)	168 (25.5)		(ref)	751 (52.7)	55 (7.3)		(ref)
High	431 (39.5)	88 (20.4)	0.052	0.75 (0.56–1.00)	675 (47.3)	26 (3.9)	<0.05	0.51 (0.31–0.82)
**Education in privacy**								
Low	679 (62.3)	180 (26.5)		(ref)	670 (47.0)	58 (8.7)		(ref)
High	411 (37.7)	76 (18.5)	<0.05	0.63 (0.47–0.85)	756 (53.0)	23 (3.0)	<0.001	0.33 (0.20–0.54)
** *Personal variables* **								
**Country**								
México	409 (37.5)	91 (22.3)		(ref)	433 (30.4)	20 (4.6)		(ref)
Chile	123 (11.3)	47 (38.2)		2.16 (1.40–3.33)	165 (11.6)	11 (6.7)		1.48 (0.69–3.15)
Spain	301 (27.6)	69 (22.9)		1.04 (0.73–1.48)	430 (30.1)	35 (8.1)		1.83 (1.04–3.22)
Perú	257 (23.6)	49 (19.1)	<0.001	0.82 (0.56–1.21)	398 (27.9)	15 (3.8)	<0.05	0.81 (0.41–1.60)
**Religiosity**								
None/Low	669 (61.4)	178 (26.6)		(ref)	977 (68.5)	66 (6.8)		(ref)
High ^c^	421 (38.6)	78 (18.5)	<0.05	0.63 (0.46–0.85)	449 (31.5)	15 (3.3)	<0.05	0.48 (0.27–0.85)
**Parents with a university education**								
No	313 (28.7)	58 (18.5)		(ref)	543 (38.1)	29 (5.3)		(ref)
Yes	777 (71.3)	198 (25.5)	<0.05	1.50 (1.08–2.09)	883 (61.9)	52 (5.9)	0.664	1.11 (0.70–1.77)
**Having a smartphone/tablet with mobile data**								
No	435 (39.9)	76 (17.5)		(ref)	428 (30.0)	23 (5.4)		(ref)
Yes	655 (60.1)	180 (27.5)	<0.001	1.79 (1.32–2.42)	998 (70.0)	58 (5.8)	0.743	1.09 (0.66–1.79)
**Having a parental content filter in the house**								
No/don’t know	866 (79.4)	219 (25.3)		(ref)	1168 (82.0)	66 (5.7)		(ref)
Yes	224 (20.5)	37 (16.5)	<0.05	0.58 (0.40–0.86)	258 (18.0)	15 (5.8)	0.918	1.03 (0.58–1.84)

^a^ *p* value for bivariate χ^2^ tests. ^b^ Univariate logistic regression odds ratios (and 95% confidence intervals) of each variable. ^c^ High religiosity refers to those with religion, with a weekly or more frequent church attendance, who consider their faith important or very important. Those who did not meet these three criteria were coded as “None/low religiosity”. Ref = reference.

**Table 3 behavsci-14-00926-t003:** Variables associated with pornography use, according to sex.

	Boys	Girls
	OR (95% CI) ^a^	OR (95% CI) ^a^
** *Family education variables* **		
**Parental warmth**		
Low	(ref)	(ref)
High	0.70 (0.50–0.98)	0.45 (0.25–0.80)
**Parental demandingness**		
Low	(ref)	(ref)
High	0.84 (0.62–1.13)	0.79 (0.47–1.33)
**Education in fortitude**		
Low	(ref)	(ref)
High	1.00 (0.70–1.42)	0.99 (0.56–1.76)
**Education in privacy**		
Low	(ref)	(ref)
High	0.76 (0.54–1.07)	0.45 (0.26–0.77)
** *Personal characteristics* **		
**Country**		
México	(ref)	(ref)
Chile	2.04 (1.31–3.20)	1.01 (0.46–2.22)
Spain	0.98 (0.68–1.42)	1.70 (0.95–3.06)
Perú	0.72 (0.48–1.08)	0.67 (0.33–1.36)
**Religiosity**		
None/Low	(ref)	(ref)
High ^b^	0.66 (0.48–0.91)	0.67 (0.36–1.24)
**Parents with a university education**		
No	(ref)	(ref)
Yes	1.70 (1.20–2.39)	1.41 (0.86–2.30)
**Having a smartphone/tablet with mobile data**		
No	(ref)	(ref)
Yes	1.80 (1.32–2.46)	0.96 (0.57–1.61)
**Having filter in the house**		
No/don’t know	(ref)	(ref)
Yes	0.69 (0.46–1.03)	1.30 (0.72–2.37)

^a^ Multiple logistic regression odds ratios (and 95% confidence intervals) of each variable, adjusted for all variables in the first column. ^b^ High religiosity refers to those with religion, with a weekly or more frequent church attendance, who consider their faith important or very important. Those who did not meet these three criteria were coded as “None/low religiosity”. Ref = reference.

## Data Availability

Available in https://doi.org/10.7910/DVN/U0ARX1, Harvard Dataverse, DRAFT VERSION.

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
