# Peer review of "Parental Competence and Pornography Use among Hispanic Adolescents"

_behavsci, 2024, doi:10.3390/bs14100926_

Round 1
Reviewer 1 Report
Comments and Suggestions for Authors
The manuscript addresses an important and under-explored area of research: the role of family education in modulating pornography consumption among Hispanic adolescents. The study's focus on a Hispanic context fills a gap in the literature, and its findings have implications for parental practices and adolescent development. However, there are several areas that require further clarification and improvement.
Major points:
Introduction:
1. The introduction section dedicates an overly long discussion to internet addiction and online pornography, with numerous paragraphs and outdated references.
2. The concepts of parenting styles, parenting competence, and parenting practices could benefit from clearer differentiation to enhance understanding. It would be helpful to define and distinguish these terms early in the introduction, providing examples or citations for each to ensure clarity.
Methodology:
1. There is a need for more detailed information on the sampling process. For example, why was the age range of the participants limited to 12-15 years old?
2. The authors' handling of missing values appears to lack clarity and consistency. For instance, regarding the Parental warmth variable, the authors state that participants who responded to fewer than half of the items were eliminated. However, if participants had missing responses that did not exceed half, how were these missing values handled?
Additionally, for the variable of the presence of parental content filters at home, participants who did not respond to this question were classified as "0" (grouped with no/don’t know), whereas for the variable of having a smartphone/tablet with mobile data, participants who did not respond were classified as "no data" (separate from the "no" group).
3. In conducting cross-national sampling and statistical analysis, ensuring measurement invariance is crucial. Although several countries in this study are Spanish-speaking, cultural backgrounds still differ. Have the authors considered the issue of measurement invariance?
4. Because the participants in this study are minors and the research topic involves pornography use, the authors need to provide a more detailed explanation of the ethical review process. For example, how was informed consent obtained? Was it from the participants themselves or their guardians?
Results:
The presentation of the results is somewhat disorganized. I suggest the authors use subheadings for each section to categorize and clearly present the research findings.
Discussion:
This study only considered the measurement of pornography use based on frequency, categorizing it roughly into two groups: yes or no. Similarly, other variables were handled by dividing them into high and low categories based on median splits, which reduces statistical power. These shortcomings should be emphasized.
Minor points:
1. The author needs to carefully review the spelling issues in the article, as there are multiple spelling errors. For example, there are two subsections titled "2.8". In the "Education in privacy" section, the author wrote: "This variable was dichotomized around the median into 'high education in fortitude' and 'low education in fortitude.'" Additionally, in Table 1, "sample" is written as "simple".
2. It's better to include sections 2.3 to 2.8 under the third-level subheading of section 2.2, labeled as 2.2.1, 2.2.2, and so on.
Comments on the Quality of English Language/
Author Response
Article: Parental competence and pornography use among Hispanic adolescents
BEHAVIOURAL SCIENCES. COMMENTS TO REVIEWERS
REVIEWER 1
Major points:
Introduction:
Comments 1: The introduction section dedicates an overly long discussion to internet addiction and online pornography, with numerous paragraphs and outdated references.
Response: Thank you for pointing this out. We agree with this comment. Therefore we removed the parts of the introduction that were repetitive, added citations (paragraph 7, 8 and 9) and also updated references (paragraph 2,4,7,8 and 9).
Comments 2: The concepts of parenting styles, parenting competence, and parenting practices could benefit from clearer differentiation to enhance understanding. It would be helpful to define and distinguish these terms early in the introduction, providing examples or citations for each to ensure clarity.
Response: Thank you for pointing this out. We agree with your comment. Therefore we have reorganized and rewrote paragraph 7, 8 and 9. We have added the definitions as suggested and updated the references. We hope this addresses your concern.
Methodology:
Comments 1: There is a need for more detailed information on the sampling process. For example, why was the age range of the participants limited to 12-15 years old?
Response: Thank you for your comment. We understand the importance of providing further clarification, and we are happy to elaborate on this point.
The age range of 12-15 years was selected for this study primarily due to methodological reasons. As mentioned in the "Questionnaire and Variables" section of the manuscript, the data collection involved three waves of questionnaires (Q13, Q15, and Q17), each focusing on different topics. The questions related to parenting styles, which are central to our analysis, were included only in the Q13 wave, which was specifically answered by adolescents aged 12-15. Consequently, data from older adolescents were not available for analysis.
We have now added a clarifying statement regarding the age range in the "Questionnaires and Variables" section to ensure this is clearly explained. Additionally, we have revised the "Participants and Sampling" section to provide further details. We hope this addresses your concern.
Comments 2: The authors' handling of missing values appears to lack clarity and consistency. For instance, regarding the Parental warmth variable, the authors state that participants who responded to fewer than half of the items were eliminated. However, if participants had missing responses that did not exceed half, how were these missing values handled?
Response: Thank you for your observation regarding the handling of missing values. We would like to clarify that for participants who completed at least half of the items in any of the four Parental Competences scales were retained. For those with fewer missing responses, the mean was calculated based on the items they answered, not the total scale, ensuring efficient use of the available data without unnecessary exclusions.
A clearer explanation of how missing values were handled for these variables has been added to the “Questionnaire and Variables” section of the manuscript.
Comments 3: Additionally, for the variable of the presence of parental content filters at home, participants who did not respond to this question were classified as "0" (grouped with no/don’t know), whereas for the variable of having a smartphone/tablet with mobile data, participants who did not respond were classified as "no data" (separate from the "no" group).
Response: Thank you for your comment. In this case, we treated the missing values differently because these variables did not represent a psychological scale or construct, but rather straightforward factual information. We opted for a more conservative strategy. Participants who did not respond were grouped with the "no/don't know" category, as our goal was to compare those who answered "yes" against a diverse comparison group (other situations), without losing subjects. This approach introduces potential heterogeneity in the comparison group, as some missing values could represent valid "yes" responses. This leads to a bias toward the null, making it harder to detect differences between groups. However, if significant differences are still found despite this heterogeneity, it suggests that the true differences may be even larger if the missing values were correctly classified. This conservative approach, therefore, strengthens the robustness of our conclusions.
We hope this explanation clarifies our rationale.
Comments 3. In conducting cross-national sampling and statistical analysis, ensuring measurement invariance is crucial. Although several countries in this study are Spanish-speaking, cultural backgrounds still differ. Have the authors considered the issue of measurement invariance?
Response: Thank you for raising the issue of measurement invariance in cross-national studies. Yes, we have considered it. The authors of the scale validation have confirmed that the instrument meets configural invariance for the specific countries included in our study. Configural invariance, or construct invariance, ensures that the same factor structure is consistent across all groups, confirming that the scale measures the same constructs in each country. Given that our study includes the same countries as the original validation, this basic level of invariance is already established.
Since the focus of our study is not on cross-cultural differences, but rather on evaluating specific relationships within the data, it was not necessary to test for higher levels of invariance (e.g., metric or scalar). We believe that ensuring configural invariance is sufficient for the objectives of our research.
Comments 4: Because the participants in this study are minors and the research topic involve pornography use, the authors need to provide a more detailed explanation of the ethical review process. For example, how was informed consent obtained? Was it from the participants themselves or their guardians?
Response: Thank you for your comment. We would like to clarify that informed consent for this study was obtained directly from the students. Before starting the questionnaire, participants were informed about the nature of the study and gave their consent by choosing to participate. Additionally, each high school was provided with the necessary documentation to inform parents about the study and obtain their consent, and we offered support to help implement their chosen protocol. The centers could choose from several established protocols depending on their usual procedures. These included implicit parental authorization (where no formal consent from parents was required), providing information to parents with implicit consent, requesting explicit opt-out, or seeking explicit parental authorization. We hope this explanation clarifies the ethical review process for our study. In consequence, we have improved the explanation regarding the consent process and ethical considerations in the “Data Collection” section of the manuscript to provide more detailed information.
Results:
Comments 1: The presentation of the results is somewhat disorganized. I suggest the authors use subheadings for each section to categorize and clearly present the research findings.
Response: Thank you for your suggestion. We have reorganized the presentation of the results and added subheadings to each section.
Discussion:
Comments 1: This study only considered the measurement of pornography use based on frequency, categorizing it roughly into two groups: yes or no. Similarly, other variables were handled by dividing them into high and low categories based on median splits, which reduces statistical power. These shortcomings should be emphasized.
Response: Done. Thank you.
Minor points:
Comments 2: The author needs to carefully review the spelling issues in the article, as there are multiple spelling errors. For example, there are two subsections titled "2.8". In the "Education in privacy" section, the author wrote: "This variable was dichotomized around the median into 'high education in fortitude' and 'low education in fortitude.'" Additionally, in Table 1, "sample" is written as "simple".
Response: Thank you for pointing this out. When we submit the article to the journal, we only put the subtitles but not the subsections, the subsections were added by the journal. Therefore, as this is a formatting for the journal, we will forward suggestions regarding subsections to the editors. The English language errors, the English language was reviewed and the errors were corrected. The writing error in the table was also corrected.
Comments 3: It's better to include sections 2.3 to 2.8 under the third-level subheading of section 2.2, labeled as 2.2.1, 2.2.2, and so on.
Response: Thank you for pointing this out. As suggested, the Dependent/result variable: use of pornography session was added to the Education in privacy session (paragraph 9 of the methods section). As we explained previously, about subsections, we only put the subtitles but not the subsections, the subsections were added by the journal. Therefore, as this is a formatting for the journal, we will forward suggestions regarding subsections to the editors.

Reviewer 2 Report
Comments and Suggestions for Authors
First of all, we must congratulate the authors of the paper. The topic is very necessary, there is great concern about the consumption of porn by our minors. Therein lies the great strength of this work, its current theme. On the other hand, it presents a great theoretical contribution and a good methodological structure. I only urge you to modify the variables and instruments to present them in a way that is easier to read and understand.
Author Response
Article: Parental competence and pornography use among Hispanic adolescents
BEHAVIOURAL SCIENCES. COMMENTS TO REVIEWERS
REVIEWER 2
Comments 1: First of all, we must congratulate the authors of the paper. The topic is very necessary, there is great concern about the consumption of porn by our minors. Therein lies the great strength of this work, its current theme. On the other hand, it presents a great theoretical contribution and a good methodological structure. I only urge you to modify the variables and instruments to present them in a way that is easier to read and understand.
Response: Thanks for your review. In response, we have revised the presentation of the variables and instruments to improve clarity and readability, ensuring that they are now easier to follow.

Reviewer 3 Report
Comments and Suggestions for Authors
The subject studied and presented in the article is very current and relevant to the health of adolescents.
However, I would like to make some comments and some suggestions for improving the article.
They are the following:
- Clarify, throughout the text and, if possible, also in the title, that parental competences are obtained through the perception of adolescents and not through interviews with parents.
- Clarify and indicate throughout the text that the objective is to study the use of digital pornography. In some places it appears digital and in others places it just appears pornography.
- The text is sometimes repetitive, which makes it tedious for the reader. Please remove the repetitions. For example, in the results, the first 5 lines are repeated with information given above.
- The second paragraph on p. 9, should be included in the discussion.
- In the discussion, the differences between boys and girls should be emphasized and should be discussed the cultural issues associated with them.
Also, in the discussion, the percentage result obtained from consumers of digital pornography should be discussed and compared with results from studies prior to the digital era.
Author Response
Article: Parental competence and pornography use among Hispanic adolescents
BEHAVIOURAL SCIENCES. COMMENTS TO REVIEWERSREVIEWER 3
The subject studied and presented in the article is very current and relevant to the health of adolescents.
However, I would like to make some comments and some suggestions for improving the article.
They are the following:
Comments 1: Clarify, throughout the text and, if possible, also in the title, that parental competences are obtained through the perception of adolescents and not through interviews with parents.
Response: Thanks for your review. In response, we have revised, reorganized and rewrote the presentation of the methodology and results. Thus, we believe that the modifications made contribute to the suggested explanations. (paragraph 2).
Comments 2: Clarify and indicate throughout the text that the objective is to study the use of digital pornography. In some places it appears digital and in others places it just appears pornography.
Response: Thank you for pointing this out. We agree with this comment. Our objective is pornography in general. In our study, the two questions we ask about pornography in the questionnaire are: "Have you seen erotic or pornographic material?" and "Have you seen erotic or pornographic material on your mobile phone?" and the number of young people who responded is very close (prevalence of pornography consumption (general): boys - 23.5% and girls - 5.7%. Prevalence of digital pornography consumption ("via mobile"): boys - 25.1%) and girls - 5.85%. Given the small difference between the two proportions, we have opted to use the general consumption variable as it provides a more comprehensive measure of pornography use. This variable is more inclusive and avoids limiting the scope to a specific medium, ensuring that we capture all forms of pornography consumption, not just digital. We hope this addresses your concern.
Comments 3: The text is sometimes repetitive, which makes it tedious for the reader. Please remove the repetitions. For example, in the results, the first 5 lines are repeated with information given above.
Response: Thank you for pointing this out. We agree with this comment. Therefore we reorganized and rewrote the presentation of the results. In the discussion section we rewrote paragraph 2,3 and 8, added citations and also updated references.We hope this addresses your concern.
Comments 4: The second paragraph on p. 9, should be included in the discussion.
Response: Thanks for your review. In response, we have revised, reorganized and rewrote the presentation of the methodology and results. Thus, we believe that the modifications made contribute to the suggested explanations. We hope this addresses your concern.
Comments 5: In the discussion, the differences between boys and girls should be emphasized and should be discussed the cultural issues associated with them.
Response: Thank you for pointing this out. We agree with this comment. Therefore we discussed differences between boys and girls and the use of pornography and the cultural aspects at paragraph 3, new citations have been added too. We hope this addresses your concern.
Comments 6: Also, in the discussion, the percentage result obtained from consumers of digital pornography should be discussed and compared with results from studies prior to the digital era.
Response: Thank you for pointing this out. We have discussed the prevalence of pornography use in general, the increasingly early use of pornography between boys and girls and the reasons why boys may be using more pornography. We have updated the references and brought in new references.We hope this addresses your concern (paragraph 2).

Round 2
Reviewer 1 Report
Comments and Suggestions for Authors
The authors have addressed the questions I raised, and the quality of the manuscript has significantly improved.
Reviewer 3 Report
Comments and Suggestions for Authors
accept in current form